# Recurrent and Multidrug-Resistant UTI Treatments in Kidney Transplant Patients: A Retrospective Study from Saudi Arabia

**DOI:** 10.3390/antibiotics14111147

**Published:** 2025-11-13

**Authors:** Khalid A. Alzahrani, Redwan Y. Mirdad, Anas T. Khogeer, Buthainah B. Alammash, Abdulfattah Y. Alhazmi, Nouf E. Alotaibi, Abdullah S. Alshammari, Abdulmalik S. Alotaibi, Mohammed A. Alnuhait

**Affiliations:** 1Makkah Health Cluster, Makkah 24237, Saudi Arabia; 2Pharmacy Department, Sukoon Long Term Care, Jeddah 21000, Saudi Arabia; 3Department of Pharmaceutical Care Services, King Fahad Hospital, Ministry of Health, Madinah 42311, Saudi Arabia; 4Pharmaceutical Practices Department, College of Pharmacy, Umm Al-Qura University, Makkah 24237, Saudi Arabia; 5Department of Clinical Pharmacy, Shaqra University, College of Pharmacy, Al-Dawadmi Campus, Al-Dawadmi 11911, Saudi Arabia

**Keywords:** urinary tract infection, kidney transplant, Saudi Arabia, antimicrobial resistance, multidrug-resistant

## Abstract

**Background:** Urinary tract infections (UTIs) are the most common infections among kidney transplant recipients, with prevalence rates ranging from 12% to 75% in studies from North America and Australia and from 4.5% to 85% in the Middle East. These infections can significantly impact graft survival and patient quality of life, increasing the risk of hospitalization, morbidity, and mortality. *Escherichia coli* is the leading cause of UTIs in transplant patients, but multidrug-resistant (MDR) pathogens are a growing concern, especially in Saudi Arabia. Several factors, including advanced age, female gender, and use of urinary catheters, contribute to post-transplant UTIs. This study focuses on the Saudi population, aiming to assess the prevalence, risk factors, and treatment strategies for recurrent and multidrug-resistant UTIs in kidney transplant recipients. **Methods:** This retrospective cohort study reviewed the medical records of kidney transplant patients at King Faisal Specialist Hospital & Research Center, Jeddah, in addition to data from King Fahad Hospital, Madinah, Saudi Arabia, between March and May 2022. Adult patients (≥18 years) who developed recurrent UTIs within two years post-transplant were included, while those with one or no UTI episode or incomplete records were excluded. **Results:** Seventy-five of 491 screened patients (15.3%) experienced recurrent UTIs, contributing to a total of 219 episodes. *Klebsiella pneumoniae* was the most frequent pathogen, isolated in 94 episodes (42.9%). Key risk factors for recurrence included complicated UTIs (OR = 4.60, *p* = 0.005), multidrug-resistant organisms (MDROs) (OR = 3.14, *p* = 0.021), and ureteric stents (OR = 4.07, *p* = 0.042). Carbapenems were primarily used for complicated UTIs, while cephalosporins and penicillins were used for uncomplicated infections. A significant post-UTI rise in serum creatinine was observed (*p* < 0.001). **Conclusions:** Recurrent UTIs predominantly caused by *K. pneumoniae* are common in kidney transplant recipients, particularly in patients over 45, with multidrug-resistant organisms, or with ureteric stents. While a direct causal link to graft loss was not established, these infections can lead to increased creatinine levels, hospitalizations, and healthcare costs and increased carbapenem use. These findings highlight the critical need for institution-specific antimicrobial stewardship programs focused on infection prevention and optimized antibiotic use to improve outcomes in this vulnerable population.

## 1. Introduction

Urinary tract infections (UTIs) are the most prevalent and enduring infections observed after kidney transplantation [1,2,3]. Research indicates that the incidence of UTIs in kidney transplant recipients varies between 12% and 75%, depending on the specific population and geographic region studied [4,5,6,7]. The occurrence of a UTI in kidney transplant patients can have far-reaching consequences, affecting both the survival of the graft and the overall quality of life for the patient. In many cases, UTIs can lead to hospitalizations, prolonged recovery periods, and an increased risk of further complications, including morbidity and mortality [8,9]. Furthermore, the management of these infections is complicated by the global rise of multidrug-resistant (MDR) uropathogens, a trend particularly pronounced in the Middle East, where high rates of extended-spectrum β-lactamase (ESBL)-producing Enterobacterales and carbapenem-resistant organisms pose a significant therapeutic challenge [10,11,12,13]. The bacterial strains most commonly implicated in post-transplant urinary tract infections include *Escherichia coli*, which accounts for 30–80% of cases. Other notable pathogens include *Klebsiella pneumoniae* (12%) and *Enterococcus* species (5%), as well as less common strains such as *Pseudomonas aeruginosa* and *Proteus* [5,6,7]. These infections can vary from uncomplicated to severe, multidrug-resistant infections that complicate treatment outcomes. Several patient-specific and transplant-related factors contribute to an increased risk of developing UTI after transplantation. Factors include advanced age, as older patients exhibit greater susceptibility, and gender, with females at a higher risk compared to males. Diabetes mellitus, receiving a kidney from a deceased donor, continued use of urinary catheters, and acute rejection episodes are additional significant risk factors [1,4,6,14]. All these characteristics can affect the frequency and severity of UTI episodes, complicating prevention and management in this demographic. This study aims to enhance the understanding of recurrent and multidrug-resistant urinary tract infections in kidney transplant patients using a retrospective analysis of patient data collected two years post-transplant among two hospitals. The main objective is to evaluate the existing therapeutic options implemented in Saudi Arabia to manage these medical conditions. The study seeks to identify the primary risk variables that increase the likelihood of UTI recurrence, possibly influencing future prevention and treatment strategies for this patient group in Saudi Arabia.

## 2. Result

The medical records of 491 renal transplant recipients were reviewed, and 75 patients (15.3%) met the inclusion criteria for recurrent UTIs. The mean age of the patients was 48.2 ± 15.8 years, with 43 (57.5%) being women. The average BMI was 24.9 ± 5.9. A significant majority (85.3%) of patients received living donor transplants (Table 1, Figure 1).

Key risk factors associated with recurrent UTIs included age over 45 (Mean = 48.2), DM (OR = 1.833, *p* = 0.039), the presence of multidrug-resistant organisms (MDRO) (OR = 3.14, 95% CI = 1.19–8.33, *p* = 0.02), ureteric stents (OR = 4.07, 95% CI = 1.05–15.67, *p* = 0.04), and complicated UTIs (OR = 4.60, 95% CI = 1.57–13.50, *p* = 0.005). Interestingly, patients with catheters were significantly less likely to experience recurrent UTIs compared to those without (OR = 0.19, 95% CI = 0.07–0.54, *p* = 0.002), and female gender was associated with lower odds of recurrence compared to male gender (OR = 0.382, 95% CI = 0.148–0.99, *p* = 0.048) (Table 2).

Logistic regression showed no significant association between baseline serum creatinine (OR = 1.006, *p* = 0.136) and white blood cell count (OR = 0.963, *p* = 0.472) with the recurrence of UTIs post-transplant. Similarly, no significant differences were found in serum creatinine levels or WBC counts between the first and second UTI episodes (*p* > 0.05). Additionally, the duration of prophylaxis (OR = 1.10, 95% CI = 0.97–1.22, *p* = 0.67) and treatment duration after the first (OR = 0.99, 95% CI = 0.86–1.14, *p* = 0.90) and second UTI episodes (OR = 1.06, 95% CI = 0.95–1.18, *p* = 0.29) did not increase the risk of recurrent UTIs. Out of 219 UTI episodes, *Klebsiella pneumoniae* was the most commonly identified organism (*n* = 94; 42.9%), followed by *Escherichia coli* (*n* = 84; 38.4%), *Pseudomonas aeruginosa* (*n* = 16; 7.3%), *Proteus mirabilis* (*n* = 7; 3.2%), and others (*n* = 18; 8.2%) (Figure 2).

The most frequently used antibiotics were carbapenems (*n* = 95; 40.8%), cephalosporins (*n* = 67; 28.2%), penicillins (*n* = 28; 12%), and fluoroquinolones (*n* = 27; 11.6%). The Wilcoxon signed-rank test revealed a significant rise in serum creatinine levels in post-transplant patients with recurrent UTIs compared to baseline levels (Z = −4.552, *p* < 0.001) (Table 3).

Serum creatinine increased in 58 patients, remained the same in three, and decreased in 14. The median post-transplant creatinine level was 97 µmol/L (IQR = 77), compared to a baseline median of 85 µmol/L (IQR = 48) (Z = −4.552, *p* < 0.001).

## 3. Discussion

Our results are consistent with other research regarding risk factors for recurrent urinary tract infections [1,4,6,7,15]. For instance, when we compared our findings with a meta-analysis encompassing around 72,600 individuals, both investigations recognized age exceeding 45 as a notable risk factor, with a mean age of 48 years in our investigation. Both studies also identified the use of ureteric stents as a major risk factor, with an odds ratio (OR) of 4.065 (95% CI: 1.054–15.670) in our study and OR  =  1.54 (95% CI: 1.16–2.06) in the meta-analysis [4]. In cases of UTIs caused by multidrug-resistant organisms (MDROs), our findings were consistent with those of a prior study that showed this risk factor nearly tripled the chances of recurrence, with an OR of 2.75 (95% CI: 1.97–3.83). Our study produced similar results, with an OR of 3.143 (95% CI: 1.185–8.334) [16]. Patients with complicated UTIs in our study had a significantly higher risk of recurrent UTIs (OR = 4.600, 95% CI: 1.567–13.502) and DM (OR = 1.833, *p* = 0.039), marking them as the primary risk factors identified. Although urinary catheters are widely recognized as a risk factor for urinary tract infections (UTIs), our analysis found that their presence associated with a reduced incidence of recurrent UTIs (OR = 0.19, 95% CI = 0.07–0.54, *p* = 0.002), potentially attributable to the transient nature of catheter use and the growing evidence supporting the benefits of early catheter removal in reducing UTI risk [17,18,19]. Some studies suggest that early removal might reduce the incidence of urinary tract infections. Also, the unexpected association between female gender and reduced recurrence risk warrants further investigation, but it may be related to unmeasured confounding factors in our cohort. Further factors, including smoking, anatomical anomalies, and infections like CMV and BK virus, did not demonstrate significant correlations with recurrent UTIs. (Table 2). All kidney transplant recipients in our study were given induction immunosuppressants, specifically thymoglobulin (ATG) (49.3%) or basiliximab (50.7%). Maintenance therapy included steroids, tacrolimus, and mycophenolate mofetil (MMF). Only 3 out of 75 patients (4%) experienced episodes of rejection. While most studies define *E. coli* as the predominant cause of UTIs post-kidney transplantation, our research identified *K. pneumoniae* as the most prevalent pathogen, accounting for 42.9% of the cases and followed by *E. coli* at 38.4%, *P. aeruginosa* at 7.3%, *P. mirabilis* at 3.2%, and other organisms representing 8.2% of the total UTI episodes. Numerous organizations, including the Infectious Disease Society of America (IDSA), the American Society of Transplantation Infectious Diseases Community of Practice, and the European Association of Urology Guidelines on Urological Infections, offer recommendations regarding the classifications of urinary tract infections (UTIs) and suitable antibiotic regimens for each [5,20,21]. In our study, we found that carbapenems (40%) were the most commonly used antibiotics to treat recurrent UTIs, followed by cephalosporins (28.8%), with an average treatment duration of around 9 days (Table 4).

The choice of antibiotic and treatment length varies based on factors such as the patient’s condition, the type of organism, whether the infection is complicated or uncomplicated, and the administration route [22,23,24,25,26]. The dominance of *Klebsiella pneumoniae* and the extensive use of carbapenems in our group is not an isolated finding but reflects a serious and well-acknowledged antimicrobial resistance (AMR) crisis within the Kingdom of Saudi Arabia and the broader Gulf region [10,11,12]. National and regional surveillance reports consistently show alarmingly high rates of extended-spectrum β-lactamase (ESBL)-producing *Enterobacterales* and carbapenem-resistant *Klebsiella pneumoniae* (CRKP) [13]. This growing resistance trend, clearly demonstrated in our cohort, compromises the efficacy of first-line antibiotics and necessitates the use of last-resort agents like carbapenems. This highlights the urgent need for institution-specific, evidence-based antimicrobial stewardship programs (ASPs) tailored for the transplant population, in line with recent national efforts [27,28,29]. Such programs should focus on the following: empiric therapy guidelines based on real-time local antibiograms, rapid diagnostic testing to enable early de-escalation from broad-spectrum drugs, and strict adherence to guidelines for appropriate treatment length. Our findings add to this vital regional discussion by pointing out the unique challenges faced by highly vulnerable kidney transplant patients. Although some studies have explored antibiotic strategies for preventing recurrent infections [30,31,32], all kidney transplant recipients in our study received UTI prophylaxis with daily trimethoprim-sulfamethoxazole for at least six months post-transplant, following KDIGO guidelines for the care of KTRs [33]. In our findings, carbapenems were predominantly used for complicated infections, while cephalosporins and penicillins were more commonly selected for uncomplicated cases. This prescribing pattern reflects the serious and well-acknowledged antimicrobial resistance (AMR) crisis within the Kingdom of Saudi Arabia and the broader Gulf region. Table 5 [34,35,36]. A Chi-Square test of independence was performed to examine the relation between antibiotic class selection and UTI complexity. The relation between these variables was significant, χ^2^(10, N = 72) = 43.462, *p* < 0.001, indicating that the choice of antibiotic was not independent of whether the infection was complicated or uncomplicated.

Numerous studies have investigated the influence of UTIs on graft function, yielding varied outcomes. Certain studies investigating the impact of urinary tract infections [37,38] indicate no correlation between urinary tract infections (UTIs) and poor transplant function or survival, but other studies recognize UTIs as a notable risk factor for unfavorable graft outcomes. Effects of recurrent urinary tract infections on graft and patient outcomes after kidney transplantation have been described [8,9]. Our investigation revealed that blood creatinine levels were elevated subsequent to urinary tract infection episodes, especially following the initial infection. A Wilcoxon signed-rank test revealed a statistically significant rise in serum creatinine levels in KTRs following UTI incidents compared to baseline levels (Z = −4.552, *p* < 0.001), with serum creatinine increasing in 58 patients (77.3%). This study has several strengths, including its conduct at two major transplant hospitals in Saudi Arabia, which enabled access to an extensive dataset and supported the examination of certain risk factors over an extended duration. The findings are consistent with existing research, and the comprehensive analysis of antibiotic use, pathogen prevalence, and their impact on serum creatinine provides valuable insights for post-transplant management. However, there are some limitations. The retrospective design of the study may introduce bias resulting from differences in hospital systems, employment, and standards over time. The limited sample size and concentration on a single region may restrict the applicability of the findings to wider populations. Furthermore, a major limitation of our analysis is the use of univariate logistic regression. While this identified several significant associations, these models are unadjusted and cannot determine whether the risk factors are independent predictors or if their effects are confounded by other variables. The wide confidence intervals for some estimates also indicate statistical uncertainty, likely due to the sample size. Therefore, the odds ratios presented should be interpreted as preliminary associations. Future studies with larger cohorts are necessary to perform multivariable analysis to adjust for potential confounders and identify true independent risk factors for recurrent UTIs in this population.

## 4. Materials and Methods

This retrospective cohort study analyzed the medical records of kidney transplant patients at King Faisal Specialist Hospital & Research Center in Jeddah, as well as data from King Fahad Hospital in Madinah, Saudi Arabia, from March to May 2022. To minimize inter-center variability and potential selection bias, a standardized set of diagnostic criteria for UTIs (detailed below) was applied uniformly to all patient records from both institutions during data collection. Furthermore, all data collectors were trained using the same protocol to ensure consistent interpretation and application of these criteria across both sites. The study was approved by the Institutional Review Boards of both hospitals (IRB 2022-22 for Jeddah and IRB 22-073 for Madinah). The inclusion criteria focused on adult patients (≥18 years) who had received a kidney transplant and developed recurrent UTIs within two years post-transplant. Exclusion criteria: Patients who experienced only one or no UTI episodes post-transplant to specifically investigate the risk factors associated with recurrent infection, a distinct clinical entity. This focus may limit the generalizability of our findings to transplant recipients with sporadic, rather than recurrent, UTIs, those with incomplete data or lost to follow-up, and patients under 18 years of age. Data collected included patient demographics, pre-transplant risk factors, transplantation-related factors, baseline lab values, immunosuppressant medications, and UTI episodes, along with their management. Definitions for classifications used in this study were based on the IDSA guidelines cited earlier in this manuscript [20]. Recurrent UTI (rUTI) was defined as two or more culture-proven UTI episodes within six months, or three or more episodes within twelve months post-transplantation, adapting the IDSA criteria for the immunocompromised transplant population. Uncomplicated UTI: Positive urine culture accompanied by urinary symptoms such as dysuria, urgency, frequency, or suprapubic pain. Complicated UTI: Systemic symptoms requiring hospital admission for intravenous antibiotic therapy. Clinical presentation: Symptoms such as fever, chills, dysuria, cloudy urine, flank pain, or tenderness in the grafted kidney. Urinalysis: Considered positive if leukocyte esterase is ≥250 and/or nitrite is positive. Risk factors recorded in the study included diabetes mellitus, smoking, structural factors (e.g., native kidney disease with urological malformations, renal cysts, calculi, benign prostate hyperplasia), neurogenic bladder, and operation-related factors (e.g., urinary catheter, ureteric stent, vesicoureteral reflux, and urinary fistula), as well as infections with CMV, BK virus, and UTI caused by *E. coli*.

We used statistical analyses to explore the relationship between various risk factors and recurrent UTIs in kidney transplant recipients. Descriptive statistics were applied to summarize the demographic and clinical characteristics of the participants. Categorical variables are reported as frequencies and percentages, while continuous variables are presented as means and standard deviations. To assess the association between risk factors and recurrent UTIs, we performed univariate logistic regression analysis. Odds ratios (ORs) and 95% confidence intervals (CIs) were calculated to estimate the strength of these relationships, with a *p*-value of less than 0.05 indicating statistical significance. For comparing paired data, we employed non-parametric tests like the Wilcoxon signed-rank test. All analyses were conducted using SPSS version 25, with a significance threshold set at *p* < 0.05 for all statistical tests.

## 5. Conclusions

In conclusion, this study identifies *Klebsiella pneumoniae* as the predominant pathogen causing recurrent UTIs in our cohort of Saudi kidney transplant recipients, necessitating frequent carbapenem use. This reflects a serious regional AMR challenge. The significant association of rUTIs with increased serum creatinine highlights their clinical impact beyond simple morbidity. These findings strongly support the implementation of tailored antimicrobial stewardship programs in transplant centers, emphasizing empiric therapy guided by local antibiograms, rapid diagnostics for de-escalation, and strict adherence to treatment duration guidelines. Future efforts must focus on optimizing infection prevention strategies and prospectively evaluating their impact on graft function, resistance rates, and patient outcomes in this vulnerable population.

## Figures and Tables

**Figure 1 antibiotics-14-01147-f001:**
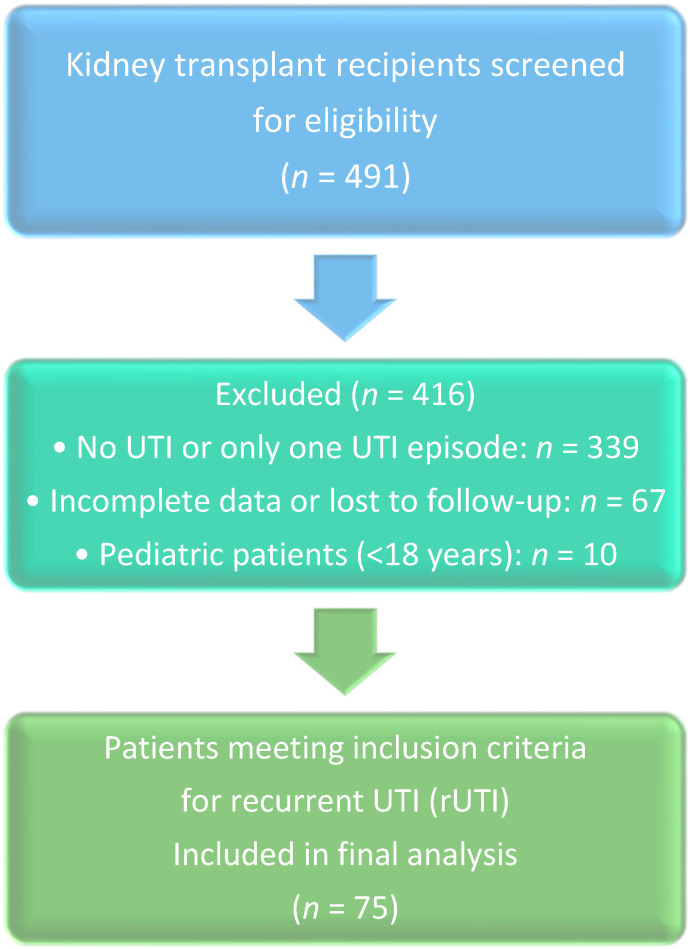
PRISMA-like flow diagram of patient selection.

**Figure 2 antibiotics-14-01147-f002:**
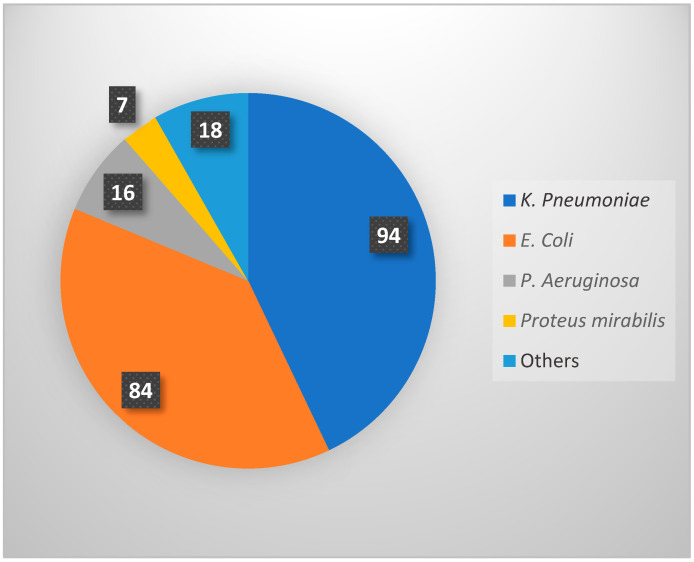
Distribution of causative microorganisms identified in 219 recurrent UTI episodes.

**Table 1 antibiotics-14-01147-t001:** Patient Eligibility and Baseline Characteristics.

Category	*n*	%
Screened Patients	491	100.0%
Included	75	15.3%
Excluded	416	84.7%
**Reason for Exclusion**		
-Single/No UTI episode	339	69.0%
-Missing data/Lost to follow-up	67	13.6%
-Pediatric age (<18 years)	10	2.0%
**Included Patients (*n* = 75)**		
**Gender**		
-Male	32	42.7%
-Female	43	57.3%
**Donor Type**		
-Living Donor	64	85.3%
-Deceased Donor	11	14.7%
**Age (years), Mean ± SD**	48.2 ± 15.8	
**BMI (kg/m^2^), Mean ± SD**	24.9 ± 5.9	

**Table 2 antibiotics-14-01147-t002:** Univariate Logistic Regression Analysis of Risk Factors Associated with Recurrent UTI.

Factor	*p*-Value	Odds Ratio (OR)	95% CI for OR
**Age (per year increase)**	0.151	1.022	0.992–1.054
**Gender (Female vs. Male)**	0.048	0.382	0.148–0.990
**Diabetes Mellitus (Yes vs. No)**	0.039	1.833	0.685–4.909
**MDRO Infection (Yes vs. No)**	0.021	3.143	1.185–8.334
**Ureteric Stent (Yes vs. No)**	0.042	4.065	1.054–15.670
**Complicated UTI (Yes vs. No)**	0.005	4.600	1.567–13.502
**Urinary Catheter (Yes vs. No)**	0.002	0.189	0.066–0.540

Abbreviations: CI, confidence interval; MDRO, multidrug-resistant organism; UTI, urinary tract infection.

**Table 3 antibiotics-14-01147-t003:** Baseline Serum Creatinine (SCr) and White Blood Cell (WBC) Counts, and Impact of First UTI Episode on SCr.

Descriptives							
			Statistic		Std. Error		
**SCr**	Mean		106.23		8.228		
	Median		85.00				
	Minimum		16				
	Maximum		544				
	Range		529				
	Interquartile Range		48				
**WBC_level**	Mean		9.16		0.553		
	Median		8.65				
	Minimum		2				
	Maximum		29				
	Range		26				
	Interquartile Range		6				
**Ranks**							
				**N**		**Mean Rank**	**Sum of Ranks**
SCr at Baseline-SCr at UTI#1		Negative Ranks		14		35.93	503.00
		Positive Ranks		58		36.64	2125.00
		Ties		3			
		Total		75			
**Test Statistics ^a^**							
		SCr at Baseline-SCr at UTI#1					
**Z**		−4.552 ^b^					
Asymp. Sig. (2-tailed)		0.000					

^a^ Wilcoxon Signed Ranks Test. ^b^ Based on negative ranks.

**Table 4 antibiotics-14-01147-t004:** Most used antibiotic classes with treatment duration.

Antibiotic Class		Used Alone (*n*)		Used in Combination (*n*)		Total Courses, *n* (%)	
Carbapenem		84		11		95 (40.8%)	
Cephalosporin		50		17		67 (28.8%)	
Penicillins		22		6		28 (12.0%)	
Fluoroquinolone		20		7		27 (11.6%)	
**Treatment Duration**		** *n* **		**%**			
<7 days		10		13.3%			
7–10 days		39		52.0%			
11–14 days		13		17.3%			
>14 days		5		6.7%			
Descriptives							
					Statistic		SE
Duration of treatment	Mean				9.7793		0.53613
	95% Confidence Interval for Mean		Lower Bound		8.7111		
			Upper Bound		10.8476		
	5% Trimmed Mean				9.2378		
	Median				9.0000		
	Variance				21.558		
	Std. Deviation				4.64304		
	Minimum				3.00		
	Maximum				37.50		
	Range				34.50		
	Interquartile Range				3.50		
	Skewness				3.349		0.277
	Kurtosis				17.162		0.548

**Table 5 antibiotics-14-01147-t005:** Association between antibiotic class selection and UTI complexity.

Chi-Square Tests	Value	df	Asymptotic Significance (2-Sided)
Pearson Chi-Square	43.462	10	<0.001
Likelihood Ratio	52.697	10	<0.001
Linear-by-Linear Association	1.413	1	0.235
N of Valid Cases	72		

## Data Availability

Upon a justified request, the corresponding author can share the data.

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
