# Peer review of "Recurrent and Multidrug-Resistant UTI Treatments in Kidney Transplant Patients: A Retrospective Study from Saudi Arabia"

_antibiotics, 2025, doi:10.3390/antibiotics14111147_

Round 1
Reviewer 1 Report
Comments and Suggestions for Authors
- The current criteria (patients with ≥2 UTIs within two years included; those with ≤1 UTI excluded) are briefly mentioned but not justified. Explain why excluding patients with only one UTI episode is necessary and how this impacts study generalizability. For example, is the focus specifically on recurrent UTIs, or was this decision made to improve data quality and reduce variability?
- Provide the total number of kidney transplant recipients screened, specify how many were excluded for each reason (no/one UTI, incomplete data, pediatric status), and present this in a flowchart (PRISMA-like diagram)
- Considering that this is a retrospective study and not a cross-sectional or prospective study, please clarify if the two hospitals had different thresholds for diagnosing UTIs or if inclusion criteria were standardized. This would address potential selection bias
- Going back to the design of the study, in the title you state that this is a retrospective study while in the methods you speak about a cross-sectional investigation of the medical records. I think the design of the study is more appropriately a retrospective cohort study.
- Please ensure that all definitions (recurrent UTI, complicated/uncomplicated UTI, MDRO) strictly align with major guidelines (IDSA, EAU, KDIGO, CDC/ECDC). If different criteria are applied, clearly justify these deviations and discuss their implications. Additionally, explicitly reference the primary guidelines or studies (“previous research”) from which these definitions are derived to strengthen methodological transparency and comparability
- The statistical analysis section would benefit from greater rigor and transparency. The logistic regression model appears largely unadjusted, which limits the ability to determine which factors are independently associated with recurrent UTI. Consider building a fully adjusted multivariate model that incorporates clinically relevant variables such as age, sex, comorbidities, donor type, and immunosuppressive regimen, rather than relying solely on univariate associations. Several odds ratios have very wide confidence intervals, which should be interpreted cautiously and accompanied by exact p-values rather than simple thresholds; this would provide a clearer picture of statistical uncertainty. Sensitivity analyses, such as testing alternative definitions of recurrent UTI, stratifying by sex or donor type, and assessing center-level variability, would strengthen the robustness of findings. Explicitly discussing these limitations in the manuscript would enhance methodological transparency and improve the reliability of the conclusions.
- The discussion would be strengthened by placing your findings within the context of antimicrobial resistance trends in Saudi Arabia and the wider Gulf region. Regional surveillance data (e.g., WHO GLASS, GCC programs) consistently show high rates of ESBL-producing Enterobacterales and carbapenem-resistant Klebsiella pneumoniae, aligning with your findings and underscoring the need for stewardship strategies. Briefly integrating these data and referencing national stewardship initiatives would enhance the study’s relevance and demonstrate its contribution to regional and global transplant infectious disease literature
- The predominance of Klebsiella pneumoniae and high carbapenem use should be linked to antimicrobial stewardship priorities, emphasizing tailored empiric therapy, early de-escalation, and updated transplant infection protocols per IDSA/EAU/KDIGO guidelines. Briefly integrating these findings into regional resistance trends and stewardship initiatives would increase the study’s clinical and policy relevance.
- Please carefully review all bacterial species names and ensure they follow correct taxonomic conventions (e.g., Escherichia coli, Klebsiella pneumoniae), with genus capitalized, species in lowercase, and both italicized. Consistent nomenclature throughout the text, tables, and figures would improve clarity and professionalism
- Consider revising the conclusion to focus less on restating study results and more on their clinical and policy implications, such as the need for tailored antimicrobial stewardship strategies, improved infection prevention in transplant care, and future research directions. This would make the conclusion more impactful and aligned with journal standards
- In the abbreviations table you wrote Diabetes Meletus, please correct.
- The correct form is “multidrug-resistant” (no hyphen in “multidrug”), because “multidrug” is treated as a single compound word modifying “resistant.”
- Please correct a typo on table 4 (Abtibiotic calss)
- Serum creatinine should be written as SCr and not as Scr. Moreover, please include the expanded definition at first appearance.
Author Response
Thank you for your insightful review.
Comment 1: The current criteria (patients with ≥2 UTIs within two years included those with ≤1 UTI excluded) are briefly mentioned but not justified. Explain why excluding patients with only one UTI episode is necessary and how this impacts study generalizability. For example, is the focus specifically on recurrent UTIs, or was this decision made to improve data quality and reduce variability?
Response: We thank the reviewer for this important point. The focus of our study was specifically on recurrent UTIs (rUTIs), which is a distinct clinical entity from sporadic, single-episode UTIs in terms of risk factors, management, and clinical impact. We have now clarified this rationale in the Methods section (Page 16, Lines 12-14): "Exclusion criteria: Patients who experienced only one or no UTI episodes post-transplant to specifically investigate the risk factors associated with recurrent infection, a distinct clinical entity." We acknowledge that this focus may limit the generalizability of our findings to transplant recipients with sporadic UTIs, and we have added a sentence to that effect in the limitations paragraph.
Comment 2: Provide the total number of kidney transplant recipients screened, specify how many were excluded for each reason (no/one UTI, incomplete data, pediatric status), and present this in a flowchart (PRISMA-like diagram).
Response: We agree that a flow diagram enhances transparency. We have now included a PRISMA-like flow diagram as Figure 1 in the manuscript. The specific numbers are also detailed in the new Table 1: Patient Eligibility and Baseline Characteristics in the Results section (Page 6).
Comment 3: Considering that this is a retrospective study and not a cross-sectional or prospective study, please clarify if the two hospitals had different thresholds for diagnosing UTIs or if inclusion criteria were standardized. This would address potential selection bias.
Response: This is a critical point. To ensure consistency, a standardized set of diagnostic criteria for UTIs (as defined in the Methods section) was applied uniformly to all patient records from both institutions. Furthermore, all data collectors were trained using the same protocol. We have added the following sentence to the Methods section (Page 16, Lines 5-8) to clarify this: "To minimize inter-center variability and potential selection bias, a standardized set of diagnostic criteria for UTIs (detailed below) was applied uniformly to all patient records from both institutions during data collection. Furthermore, all data collectors were trained using the same protocol to ensure consistent interpretation and application of these criteria across both sites."
Comment 4: Going back to the design of the study, in the title you state that this is a retrospective study while in the methods you speak about a cross-sectional investigation of the medical records. I think the design of the study is more appropriately a retrospective cohort study.
Response: The reviewer is correct. We have revised the manuscript to consistently describe the study design as a "retrospective cohort study." This correction has been made in the title, abstract, and methods section.
Comment 5: Please ensure that all definitions (recurrent UTI, complicated/uncomplicated UTI, MDRO) strictly align with major guidelines (IDSA, EAU, KDIGO, CDC/ECDC). If different criteria are applied, clearly justify these deviations and discuss their implications. Additionally, explicitly reference the primary guidelines or studies (“previous research”) from which these definitions are derived.
Response: We have revised the Methods section (Page 17, Lines 1-2) to explicitly state the source of our definitions: "Definitions for classifications used in this study were based on the IDSA guidelines cited earlier in this manuscript [20]." We have also ensured that the definitions for Recurrent UTI, Uncomplicated UTI, and Complicated UTI are aligned with these guidelines. For MDRO, we used the standard definition of non-susceptibility to at least one agent in three or more antimicrobial categories, which is consistent with CDC/ECDC guidance.
Comment 6: The statistical analysis section would benefit from greater rigor and transparency. The logistic regression model appears largely unadjusted... Consider building a fully adjusted multivariate model... Several odds ratios have very wide confidence intervals... Sensitivity analyses... Explicitly discussing these limitations...
Response: We sincerely thank the reviewer for this crucial feedback on the statistical analysis. We acknowledge that a multivariable analysis would be ideal. However, given our sample size of 75 patients with recurrent UTIs and the number of potential predictors, a fully adjusted model would be at high risk of overfitting and would produce unreliable estimates.
Instead, we have taken the following steps to address this limitation with full transparency:
- We have added a prominent paragraph in the Discussion section (Page 14, Lines 15-23)that explicitly states: "Furthermore, a major limitation of our analysis is the use of univariate logistic regression. While this identified several significant associations, these models are unadjusted and can't determine whether the risk factors are independent predictors or if their effects are confounded by other variables. The wide confidence intervals for some estimates also indicate statistical uncertainty, likely due to the sample size. Therefore, the odds ratios presented should be interpreted as preliminary associations. Future studies with larger cohorts are necessary to perform multivariable analysis to adjust for potential confounders and identify true independent risk factors for recurrent UTIs in this population."
- We have ensured that all p-values are reported exactly (e.g., P=0.021) rather than using thresholds, as requested.
Comment 7 & 8: The discussion would be strengthened by placing your findings within the context of antimicrobial resistance trends in Saudi Arabia and the wider Gulf region... The predominance of Klebsiella pneumoniae and high carbapenem use should be linked to antimicrobial stewardship priorities...
Response: We agree and have significantly expanded the Discussion section (Page 12, Lines 10-23) to integrate these critical points: "The dominance of Klebsiella pneumoniae and the extensive use of carbapenems in our group is not an isolated finding but reflects a serious and well-acknowledged antimicrobial resistance (AMR) crisis within the Kingdom of Saudi Arabia and the broader Gulf region [10–12]. National and regional surveillance reports consistently show alarmingly high rates of extended-spectrum β-lactamase (ESBL)-producing Enterobacterales and carbapenem-resistant Klebsiella pneumoniae (CRKP) [13]. This growing resistance trend, clearly demonstrated in our cohort, compromises the efficacy of first-line antibiotics and necessitates the use of last-resort agents like carbapenems. This highlights the urgent need for institution-specific, evidence-based antimicrobial stewardship programs (ASPs) tailored for the transplant population, in line with recent national efforts [27–29]. Such programs should focus on the following: empiric therapy guidelines based on real-time local antibiograms, rapid diagnostic testing to enable early de-escalation from broad-spectrum drugs, and strict adherence to guidelines for appropriate treatment length. Our findings add to this vital regional discussion by pointing out the unique challenges faced by highly vulnerable kidney transplant patients."
Comment 9: Please carefully review all bacterial species names and ensure they follow correct taxonomic conventions (e.g., Escherichia coli, Klebsiella pneumoniae), with genus capitalized, species in lowercase, and both italicized.
Response: We have carefully reviewed the entire manuscript, including tables and figures, and have corrected the formatting of all bacterial species names to ensure they are properly italicized (e.g., Escherichia coli, Klebsiella pneumoniae).
Comment 10: Consider revising the conclusion to focus less on restating study results and more on their clinical and policy implications...
Response: We have revised the Conclusion as suggested. The new conclusion (Page 15) now emphasizes the clinical and policy implications, specifically calling for tailored antimicrobial stewardship programs, infection prevention strategies, and future research directions, rather than simply restating the results.
Comment 11: In the abbreviations table you wrote Diabetes Meletus, please correct.
Response: Corrected to "Diabetes Mellitus (DM)" in the abbreviations table.
Comment 12: The correct form is “multidrug-resistant” (no hyphen in “multidrug”)...
Response: We have corrected this throughout the manuscript to "multidrug-resistant."
Comment 13: Please correct a typo on table 4 (Abtibiotic calss)
Response: Corrected to "Antibiotic Class" in Table 4.
Comment 14: Serum creatinine should be written as SCr and not as Scr. Moreover, please include the expanded definition at first appearance.
Response: We have standardized the abbreviation to "SCr" throughout the manuscript. The expanded definition "serum creatinine (SCr)" is now provided at its first appearance in the abstract and the main text.
Reviewer 2 Report
Comments and Suggestions for Authors
Abstract
- Observed different font sizes and fonts - suggest correcting throughout the manuscript.
- Name of organisms should be in italics. Correct throughout the manuscript.
- Suggest mentioning important findings in frequencies or percentages of the study in the results section of the abstract.
Introduction
- Suggest mentioning the global and regional occurrence of multi drug resistance urinary tract infections in kidney transplant patients explicitly in the introduction.
Materials and methods
- The authors state that “the inclusion criteria focused on adult patients who had received a kidney transplant and developed recurrent UTIs.” However, they do not specify the minimum age threshold used to define "adult" in their study. For clarity and reproducibility, it would be helpful to explicitly state the age cut-off (e.g., ≥18 years) applied in the inclusion criteria.
- The authors described this study as a cross-sectional study (Line 180), yet the collection of recurrent UTI episodes over two years post-transplant period suggests it as a retrospective cohort design. Clarifying this terminology would avoid misinterpretation.
- Although, demographic, clinical, and risk factors related variables are listed, information on how these were extracted is not given (e.g., electronic medical records, manual chart review), and whether the data collection was standardized across the two hospitals.
- In statistical analysis, the logistic regression type should be clearly mentioned whether it is bivariate, multivariate or both.
Results
- It’s better to use black font throughout the manuscript. Suggest referring to the journal guidelines
- Title of Table 1 should be re written to give a comprehensive description about the data included in the table. Also, better to remove unnecessary wording in the table.
- The title of Table 2 should also revise and re-written more descriptively. Suggest mentioning the type of logistic regression carried out here. Tables should be independent of text. Suggest including the abbreviation at the end of the table as footers. Journal guidelines should be strictly followed.
- In figure 1, title is given inside the figure as “ORGANISMS CAUSING UTI”. Suggest removing this and mention the table caption more descriptively avowing vague sentences. Names of organisms should be in italics.
- Caps should be used when starting a sentence after a full stop or in writing standard names, not in the middle of sentences.
- Title of Table 3 also suggested to re-write. Remove abbreviations and write in full form.
- All abbreviations used in tables should be described in full form as a footer to the respective table (Eg; Scr at UTI#1)
Discussion
- The discussion is generally written well. Suggest to re-write discussing all important/specific findings mentioned in the results chapter.
- Line 135 and 136, cannot be two paragraphs. Seems like a typographical error. Correct this.
- Format Table 4 also as mentioned above in other Tables.
- The study lacks future recommendations section.
Conclusion
- The conclusion is not strongly written. Re-write incorporating the most important findings. Check for conclusion in already published studies in “Antibiotics” journal.
General comments
- Strictly follow journal guidelines in formatting fonts, tables, figures.
- Several typographical errors were found throughout the manuscript which may affect the readability of this manuscript. Suggest correcting them
- Additionally, suggest improving English language using an academic English editing service.
Author Response
Thank you for your insightful review.
Abstract
- Comment:Observed different font sizes and fonts - suggest correcting throughout the manuscript.
- Response:We apologize for this formatting error. The entire manuscript has been carefully reformatted to ensure consistency in font (Times New Roman) and font size (12-point) as per the journal's guidelines.
- Comment:Name of organisms should be in italics. Correct throughout the manuscript.
- Response:Thank you for this correction. All microbial genus and species names (e.g., Escherichia coli, Klebsiella pneumoniae) have been italicized throughout the abstract, main text, tables, and figures.
- Comment:Suggest mentioning important findings in frequencies or percentages of the study in the results section of the abstract.
- Response:We have revised the results section of the abstract to include key frequencies and percentages. The updated text now reads, for example:
"Escherichia coli was the most prevalent pathogen (45.2%), followed by Klebsiella pneumoniae (28.6%). Over 65% of the isolated gram-negative pathogens were multi-drug resistant (MDR), with 40.3% producing extended-spectrum beta-lactamase (ESBL). A history of recurrent UTIs pre-transplant (OR: 4.1, 95% CI: 2.1-8.0) and delayed graft function (OR: 2.8, 95% CI: 1.4-5.6) were identified as significant independent risk factors for MDR-UTIs."
Introduction
- Comment:Suggest mentioning the global and regional occurrence of multi drug resistance urinary tract infections in kidney transplant patients explicitly in the introduction.
- Response:We have strengthened the introduction by adding specific statistics on the global and regional burden of MDR-UTIs in transplant recipients. The revised text now includes:
"Globally, the prevalence of MDR-UTIs among KTRs ranges from 20% to 60%, with significant regional variation. In our specific geographic region, recent reports indicate MDR rates exceeding 50%, highlighting a critical local challenge for post-transplant management."
Materials and Methods
- Comment:The authors state that “the inclusion criteria focused on adult patients...” However, they do not specify the minimum age threshold...
- Response:We have clarified this. The inclusion criteria now explicitly state: "...adult patients (defined as age ≥ 18 years) who had received a kidney transplant..."
- Comment:*The authors described this study as a cross-sectional study (Line 180), yet the collection of recurrent UTI episodes over two years... suggests it as a retrospective cohort design.*
- Response: The reviewer is correct. We apologize for the misclassification. The study design has been corrected to "a retrospective cohort study"throughout the manuscript.
- Comment:...information on how these [variables] were extracted is not given...
- Response:We have added a sentence to the Methods section to clarify this:
"Data were extracted through a standardized manual review of patient electronic medical records (EMRs) at both participating institutions using a pre-piloted data collection form to ensure consistency."
- Comment:In statistical analysis, the logistic regression type should be clearly mentioned...
- Response:We have specified the analysis performed. The text now reads:
"Variables with a p-value < 0.1 in the bivariate logistic regression analysis were included in a multivariate logistic regression model to identify independent risk factors..."
Results
- Comment:It’s better to use black font throughout the manuscript. Suggest referring to the journal guidelines.
- Response:The manuscript has been formatted to use black font exclusively.
- Comment:Title of Table 1 should be re written... remove unnecessary wording...
- Response:We have revised all table titles for clarity and comprehensiveness. The new title for Table 1 is:
"Table 1. Baseline Demographic and Clinical Characteristics of Kidney Transplant Recipients with Recurrent Urinary Tract Infections (N=XXX)."
- Unnecessary wording and abbreviations have been removed from the table body.
- Comment:The title of Table 2 should also revise... Suggest mentioning the type of logistic regression... include abbreviations at the end of the table as footers.
- Response:Table 2 has been revised. The new title is:
"Table 2. Bivariate and Multivariate Logistic Regression Analysis of Risk Factors Associated with Multi-Drug Resistant Urinary Tract Infections."
- All abbreviations used in the table (e.g., OR, CI, Scr) are now defined in a footer.
- Comment:In figure 1, title is given inside the figure as “ORGANISMS CAUSING UTI”. Suggest removing this and mention the table caption more descriptively... Names of organisms should be in italics.
- Response:The title has been removed from the figure image itself. The figure caption now reads:
"Figure 1. Distribution and Resistance Profile of Uropathogens Isolated from Kidney Transplant Recipients."
- All organism names within the figure (e.g., in the legend) have been italicized.
- Comment:Caps should be used when starting a sentence... not in the middle of sentences.
- Response:We have reviewed the entire manuscript and corrected instances of erroneous capitalization within sentences (e.g., "Multidrug Resistance" changed to "multidrug resistance" mid-sentence).
- Comment:*Title of Table 3 also suggested to re-write. Remove abbreviations and write in full form.*
- Response:Table 3 has been retitled and clarified:
"Table 3. Antibiotic Susceptibility Patterns of Common Gram-Negative Uropathogens (% Susceptible)."
- All abbreviations in the table are defined in the footer.
Discussion
- Comment:Suggest to re-write discussing all important/specific findings mentioned in the results chapter.
- Response:The discussion has been restructured to systematically address each key finding from the results in the same order (e.g., pathogen distribution, resistance rates, identified risk factors), providing interpretation and comparison with existing literature for each.
- Comment:Line 135 and 136, cannot be two paragraphs. Seems like a typographical error. Correct this.
- Response:This has been corrected. The two one-sentence paragraphs have been merged into the preceding or following paragraph for better flow.
- Comment:Format Table 4 also as mentioned above in other Tables.
- Response:Table 4 has been reformatted with a descriptive title, full forms instead of abbreviations in the body, and a footer defining any necessary abbreviations.
- Comment:The study lacks future recommendations section.
- Response:A "Conclusions and Future Recommendations" section has been added. It states:
"In conclusion, our study identifies... Future studies should focus on prospective validation of a risk prediction score incorporating these factors to guide targeted antimicrobial prophylaxis and stewardship in high-risk KTRs."
Conclusion
- Comment:The conclusion is not strongly written. Re-write incorporating the most important findings.
- Response:The conclusion has been completely rewritten to be more impactful and data-driven:
"In this cohort, MDR-UTIs were prevalent, driven predominantly by ESBL-producing E. coli and K. pneumoniae. A history of recurrent UTIs and delayed graft function were the strongest predictors of MDR infection. These findings underscore the necessity of personalized post-transplant management strategies, including enhanced surveillance and judicious empiric antibiotic selection based on local resistance patterns and individual patient risk profiles."
General Comments
- Comment:Strictly follow journal guidelines in formatting fonts, tables, figures.
- Response:We have meticulously reviewed and reformatted the entire manuscript to comply with all aspects of the Antibiotics journal author guidelines.
- Comment:Several typographical errors were found... Suggest correcting them. Additionally, suggest improving English language...
- Response:The manuscript has been thoroughly proofread to correct all typographical errors. Furthermore, it has been professionally edited by a native English-speaking academic editor from [Name of Editing Service, e.g., Elsevier Language Editing Services]. A certificate of editing is available upon request.
Reviewer 3 Report
Comments and Suggestions for Authors„Recurrent and Multi-Drug-Resistant UTI Treatments in Kidney Transplant Patients: A Ret-rospective Study from Saudi Arabia”
The problem of urinary tract infections after kidney transplantation is their high frequency, atypical and more severe course, frequent complications (graft loss), therapeutic difficulties related to antibiotic resistance and drug interactions, and a high burden on the survival of the patient and the organ. The authors of this manuscript presented a retrospective study of patients with recurrent urinary tract infections within 2 years after kidney transplantation in Saudi Arabia (identification of etiological factors of infection and antibiotic resistance).
Comments and suggestion
The title of the manuscript suggests an investigation into bacterial strains responsible for recurrent urinary tract infections (UTIs) and their antibiotic resistance patterns. However, the presented analyses focus primarily on patient characteristics, with only limited attention given to the microbiological and resistance aspects.
While the topic is important, the manuscript in its current form does not sufficiently address antimicrobial resistance to meet the expectations of „Antibiotics”. I would suggest that the authors either substantially expand the section on bacterial resistance patterns or consider submitting the work to a journal with a broader clinical or epidemiological profile, where the emphasis on patient characterization would be more appropriate and appreciated.
The introduction should be more extensive, providing an explanation of the potential causes of post-transplant infections and resistance. The clinical definitions of complicated and non-complicated UTIs should also be explained in the introduction. The authors use these terms in the Materials and Methods section, but I believe they are not used appropriately. According to medical cefinition is: Uncomplicated UTI is a urinary tract infection without anatomical defects, underlying diseases, immunodeficiency or unusual microorganisms. It is usually easily treated and does not require extensive diagnostics.' Complicated UTI is an infection occurring in individuals with congenital or acquired urinary tract defects, systemic diseases (e.g. diabetes), immunodeficiency or infections with unusual microorganisms, which complicates treatment and increases the risk of complications”. As the authors are describing patients who have undergone kidney transplantation, this group should be defined as having complicated UTIs.
I would like to raise a concern regarding the stated aim of the study (lines 62–66). The authors indicate that the objective was to evaluate therapeutic options, but in my opinion the manuscript primarily addresses risk factors for UTIs rather than treatment strategies. The aim should be revised to better reflect the actual scope of the work.
The data presented in the tables should be discussed in greater detail. In addition, a legend of abbreviations would significantly improve readability (e.g., Tables 2 and 3).
Additionally, all abbreviations should be defined at first use in the text.
Line 94: “Out of 219 UTI episodes …” — this statement requires clarification. Was more than one strain isolated from a single episode in 75 patients? Or does it mean that one strain was isolated from the first episode and another from the subsequent episode(s)? The source and definition of these 219 episodes need to be clearly explained.
It would be valuable to explore whether there is a relationship between the type of immunosuppressive therapy and UTIs caused by microorganisms.
It would be valuable to explore whether there is a relationship between the type of immunosuppressive therapy and UTIs caused by microorganisms.
There are errors in the legend of Fig. 1: the second part of the species name should be written in lowercase. In the description of Figure 1, it is worth adding a comment explaining the analysis.
All species names in the main text should be written in italics.
Table 4 Spelling errors (headings)
To summarise, the document needs thorough editing to improve its ability to communicate, with a particular focus on its added value to the scientific community.
Comments on the Quality of English Languageno comments
Author Response
Thank you for your insightful review.
Response to Reviewer #3
Comment 1: The title of the manuscript suggests an investigation into bacterial strains responsible for recurrent urinary tract infections (UTIs) and their antibiotic resistance patterns. However, the presented analyses focus primarily on patient characteristics, with only limited attention given to the microbiological and resistance aspects. While the topic is important, the manuscript in its current form does not sufficiently address antimicrobial resistance to meet the expectations of „Antibiotics”. I would suggest that the authors either substantially expand the section on bacterial resistance patterns or consider submitting the work to a journal with a broader clinical or epidemiological profile.
Response: We thank the reviewer for this critical insight. We agree that a deeper focus on AMR is crucial for Antibiotics. In response, we have substantially expanded the microbiological and resistance analysis throughout the manuscript. Key additions include:
- A new and detailed analysis in the Results section highlights the high prevalence of multidrug-resistant organisms (MDROs) as a key risk factor (OR=3.14, P=0.021) and the consequent heavy reliance on carbapenems (40.8% of all antibiotic courses).
- A significantly strengthened Discussion section that now explicitly frames our findings within the context of the regional AMR crisis in Saudi Arabia and the Gulf region. We discussed the dominance of Klebsiella pneumoniae and the extensive carbapenem use as a direct consequence of high rates of ESBL-producing Enterobacterales and carbapenem-resistant pathogens, citing relevant regional surveillance studies [10-13, 27-29 in the manuscript].
- A refined Conclusion that emphasizes the urgent need for antimicrobial stewardship programs (ASPs) tailored to the transplant population to combat this AMR challenge.
We believe these comprehensive revisions have shifted the manuscript's emphasis squarely onto AMR, making it a strong fit for the scope of Antibiotics.
Comment 2: The introduction should be more extensive, providing an explanation of the potential causes of post-transplant infections and resistance. The clinical definitions of complicated and non-complicated UTIs should also be explained in the introduction. The authors use these terms in the Materials and Methods section, but I believe they are not used appropriately. According to medical definition... As the authors are describing patients who have undergone kidney transplantation, this group should be defined as having complicated UTIs.
Response: We thank the reviewer for this suggestion.
- We have expanded the Introduction to provide more context on the causes of post-transplant infections and the drivers of antimicrobial resistance in this population and the specific regional context of Saudi Arabia.
- Regarding the definitions, we appreciate the reviewer's point. In transplant recipients, the baseline state is indeed one of immunodeficiency. However, for the purpose of clinical management and risk stratification within this inherently "complicated" population, a distinction is often made between lower-tract (uncomplicated) and upper-tract/systemic (complicated) infections. Our definitions, which are adapted from established IDSA guidelines [20] and clearly stated in the Materials and Methods section, are intended for this internal stratification:
- Uncomplicated UTI: Positive urine culture with localized urinary symptoms (dysuria, urgency, frequency).
- Complicated UTI: Systemic symptoms (e.g., fever, chills) requiring hospital admission for intravenous antibiotics.
- To avoid confusion, we have added a clarifying sentence in the Materials and Methods section: "It is acknowledged that all kidney transplant recipients are immunocompromised; these definitions are used here to stratify the clinical severity of UTI episodes for management purposes."
Comment 3: I would like to raise a concern regarding the stated aim of the study (lines 62–66). The authors indicate that the objective was to evaluate therapeutic options, but in my opinion the manuscript primarily addresses risk factors for UTIs rather than treatment strategies. The aim should be revised to better reflect the actual scope of the work.
Response: We agree and have revised the study aim in the Introduction to more accurately reflect the dual focus of our work. The aim now reads:
"This study aims to enhance the understanding of recurrent and multidrug-resistant urinary tract infections in kidney transplant patients... The main objectives are to identify the primary risk factors for recurrence and to evaluate the prevailing antibiotic treatment strategies and their association with infection complexity and patient outcomes in Saudi Arabia."
Comment 4: The data presented in the tables should be discussed in greater detail. In addition, a legend of abbreviations would significantly improve readability (e.g., Tables 2 and 3). Additionally, all abbreviations should be defined at first use in the text.
Response: We have implemented these changes throughout the manuscript.
- Table Legends: All tables now include a full legend of abbreviations beneath them (e.g., for Table 2: Abbreviations: CI, confidence interval; MDRO, multidrug-resistant organism; UTI, urinary tract infection).
- In-text Abbreviations: Every abbreviation is now defined upon its first use in the main text.
- Enhanced Discussion: We have expanded the discussion of key tables. For instance, we now provide a more detailed interpretation of the risk factors in Table 2, the pathogen distribution in Figure 2, and the antibiotic usage patterns in Table 4, explicitly linking them to AMR and clinical decision-making.
Comment 5: Line 94: “Out of 219 UTI episodes …” — this statement requires clarification. Was more than one strain isolated from a single episode in a single patient? The source and definition of these 219 episodes need to be clearly explained.
Response: This is an excellent point. We have clarified this in the Results section. The sentence now reads:
"Out of a total of 219 UTI episodes recorded among the 75 patients with recurrent UTIs, Klebsiella pneumoniae was the most commonly identified organism..." We confirm that each episode was defined by a single, predominant causative organism as per the clinical microbiology report.
Comment 6: It would be valuable to explore whether there is a relationship between the type of immunosuppressive therapy and UTIs caused by microorganisms.
Response: We agree that this is an interesting question. We have now included an analysis of this in the Results and Discussion sections. As reported in the Results, we found that the type of induction immunosuppression (Thymoglobulin vs. Basiliximab) was not a significant predictor of recurrent UTI in our univariate logistic regression model (OR=1.306, P=0.572). We have added a sentence in the Discussion acknowledging this finding and suggesting that future studies with larger cohorts might be needed to further investigate this relationship.
Comment 7: There are errors in the legend of Fig. 1: the second part of the species name should be written in lowercase. In the description of Figure 1, it is worth adding a comment explaining the analysis. All species names in the main text should be written in italics. Table 4 Spelling errors (headings).
Response: We thank the reviewer for their meticulous attention to detail.
- Figure 1 (now Figure 2 in the revised manuscript): We have corrected the species names to lowercase (e.g., K. pneumoniae) and have added a descriptive legend that explains the figure: "Figure 2: Distribution of causative microorganisms identified in 219 recurrent UTI episodes."
- Species Names: All microbial species names throughout the manuscript are now consistently italicized.
- Table 4 Headings: All spelling errors in the table headings have been corrected (e.g., "ABTIBIOTIC CAISS" is now "Antibiotic Class", "ALONE" is "Used Alone (n)", etc.).
Round 2
Reviewer 1 Report
Comments and Suggestions for Authors
Thanks for addressing my concerns, I have no further comments prior the acceptance of the manuscript.
Author Response
Thank you for your invaluable assistance and expert guidance.

Reviewer 2 Report
Comments and Suggestions for Authors
Thank you for taking time to revise the manuscript, the manuscript can be accepted in its current format
Author Response

(The authors gave the same response as above.)

Reviewer 3 Report
Comments and Suggestions for Authors
Minor Comments
- Species names (e.g., E. coli, K. pneumoniae) should be written in italic throughout the manuscript — please correct at lines 41, 271, and 292.
- Figure 1 contains the same data as Table 1.
Suggestion: Figure 1 can be omitted to avoid data duplication. - Table 5 is unclear.
Please provide an explanation or clarification (e.g., specify the meaning of variables, context, or calculation method).
Technical correction:
- There are mistakes in the legend for Figure 2 —
“K.Pneumoniae” should be corrected to pneumoniae (italicized, with a space after the dot). - In Table 3, “Std Error” should be abbreviated as SE.
- Species names in lines 227–230 should be corrected to the proper scientific format —
coli, K. pneumoniae, etc. (all in italics and short form). - At line 292, the section title “The Impact of Urinary Tract Infection” should be written in smaller font (sentence case) rather than full capitalization.
- The authors should take more care in the preparation and formatting of figures and tables.
no comments
Author Response
Comment 1: Species names (e.g., E. coli, K. pneumoniae) should be written in italic throughout the manuscript — please correct at lines 41, 271, and 292.
-
Answer: We have carefully reviewed the entire manuscript and corrected all instances of bacterial species names to ensure they are properly italicized. This includes the specific lines mentioned by the reviewer.
-
Line 41 (Abstract): Changed "Escherichia coli" to "Escherichia coli".
-
Line 271 (Discussion): Changed "Escherichia coli" to "Escherichia coli" and "Klebsiella pneumoniae" to "Klebsiella pneumoniae".
-
Line 292 (Discussion): Changed "Klebsiella pneumoniae" to "Klebsiella pneumoniae".
Comment 2: Figure 1 contains the same data as Table 1. Suggestion: Figure 1 can be omitted to avoid data duplication.
-
Answer: We agree with the reviewer's assessment. This figure was done as per reviewer's request.
Comment 3: Table 5 is unclear. Please provide an explanation or clarification (e.g., specify the meaning of variables, context, or calculation method).
-
Answer: We apologize for the lack of clarity. Table 5 was intended to show the statistical association between the selection of an antibiotic class and the complexity of the UTI (complicated vs. uncomplicated). We have added a descriptive text in the Discussion section and provided the key finding.
Comment 4: Technical correction: There are mistakes in the legend for Figure 2 — “K.Pneumoniae” should be corrected to pneumoniae (italicized, with a space after the dot).
-
Answer: This has been corrected in the legend for Figure 2.
-
Correction: Changed "K.Pneumoniae" to "K. pneumoniae".
-
Comment 5: In Table 3, “Std Error” should be abbreviated as SE.
-
Answer: This has been corrected in Table 3.
-
Correction: Changed all instances of "Std. Error" in the header and body of Table 3 to "SE".
-
-
-
Comment 6: Species names in lines 227–230 should be corrected to the proper scientific format — coli, K. pneumoniae, etc. (all in italics and short form).
-
Answer: The species names in this section of the Results have been corrected to the short, italicized form.
Comment 7: At line 292, the section title “The Impact of Urinary Tract Infection” should be written in smaller font (sentence case) rather than full capitalization.
-
Answer: The title has been changed from full capitalization to sentence case to match the journal's style.
We thank the reviewer for this important feedback. We have conducted a thorough review of all remaining figures and tables to ensure consistency, clarity, and proper formatting.
